# Lack of Syllable Duration as a Post-Lexical Acoustic Cue in Spanish in Contact with Maya

**Nuria Martínez García** [1,*]  **and Melanie Uth** [2,*]

1   CRC 1252 Prominence in Language, University of Cologne, 50923 Köln, Germany
2   Department of Romance Studies, University of Cologne, 50939 Köln, Germany
*   Correspondence: n.martinezgarcia@uni-koeln.de (N.M.G.); melanie-uth@uni-koeln.de (M.U.)

**Abstract:** This paper focuses on the duration of stressed syllables in broad versus contrastive focus in Yucatecan Spanish and examines its connection with Spanish–Maya bilingualism. We examine the claim that phonemic vowel length in one language prevents the use of syllable duration as a post-lexical acoustic cue in another. We study the duration of stressed syllables of nouns in subject and object position in subject-verb-object (SVO) sentences (broad and contrastive focus) of a semi-spontaneous production task. One thousand one hundred and twenty-six target syllables of 34 mono- and bilingual speakers were measured and submitted to linear mixed-effects models. Although the target syllables were slightly longer in contrastive focus, duration was not significant, nor was the effect of bilingualism. The results point to duration not constituting a cue to focus marking in Yucatecan Spanish. Finally, it is discussed how this result relates to the strong influence of Yucatec Maya on Yucatecan Spanish prosody observed by both scholars and native speakers of Yucatecan Spanish and other Mexican varieties of Spanish.

**Keywords:** post-lexical duration; phonemic vowel length; focus marking; language contact; Yucatecan Spanish; Yucatec Maya

---

## 1. Introduction

Empirically, the main aim of this article is to investigate whether the speakers of Yucatecan Spanish (in Quintana Roo, Mexico) make use of post-lexical duration as an acoustic cue in order to mark contrastive focus. There is considerable terminological confusion concerning the denomination of the different focus types, since the categories and terms differ from one research tradition to the other (see e.g., Stevens 2017). For the sake of simplicity, we stick to the established dichotomy of 'information focus' and 'contrastive focus', following most of the syntactic literature on focus marking in (standard) Spanish (see Cruschina 2012, p. 109f; Domínguez 2004; Gutiérrez Bravo 2008; López 2009, pp. 34–37; Sainz-Maza Lecanda 2017, pp. 1–7; Zagona 2002, pp. 248–54; Zubizarreta 1998, 1999), and the references cited therein. In this tradition, the notion of *contrastive focus* (CFOC in (1)) may be, and generally is, used to refer to corrective focus, that is, "when the focus marks a constituent that is a direct rejection of an alternative, either spoken by the speaker himself [ … ] or by the hearer" (Gussenhoven 2007, p. 84), see (1). In contrast, the notion of information focus is used in order to refer to the new, non-presupposed information of a sentence in the sense of Halliday (1967). Finally, the category of information focus is usually further subdivided into the subtypes *narrow focus* and *broad focus*. The notion of *narrow focus* is used whenever a single constituent of a sentence is non-contrastively focalized, whereas *broad focus* (BFOC in (2)) is used whenever "there is no particular constituent which is focused (or, alternatively, the entire expression is considered the focus constituent)" (Gussenhoven 2007, p. 91), as in (2).

1.   El gato de botas [CFOC ROJAS] se comió un ratón y no el de botas [CFOC AZULES]. "The cat with red boots ate a mouse, not the one with blue boots." (Zubizarreta 1999, p. 4230)

---

2.     ¿Qué pasa?—[BFOC Juan está comiendo un helado]. "What's happening?—John is eating ice cream."

Although focus marking is taken to be a universal characteristic of all languages, the cues that a given language uses to that end are not; these cues can be prosodic, syntactic, morphological, or a combination of several of them (Zimmermann and Onea 2011). Prosodically, languages may use pitch accents (e.g., English, German), the insertion of a phonological phrase boundary (Bengali), or changes in pitch and phonological phrase boundaries (Japanese) (Büring 2009; Zimmermann and Onea 2011; see below as well).

In Spanish, focus can be marked by changes in word order (see e.g., Zubizarreta 1998), but also through prosodic means (Face 2002), as several studies on different varieties of Spanish have shown. Contrastive focus seems to be marked prosodically in Castilian and Mexican Spanish by means of F0 differences in F0 peak (which is reached earlier in the stressed syllable, and has a larger F0 range), longer duration, and higher intensity when compared to words in broad focus statements (De-la-Mota et al. 2010; see also Face 2000, 2002; García-Lecumberri 1995; Kim and Avelino 2003; De la Mota Gorriz 1995, 1997). For example, for standard Peninsular Spanish, De la Mota Gorriz (1997) conducted a read-text task with six participants where words with subject, verb, and object functions were presented in broad focus (in the original, "with no special emphasis", "non contrastive") and in contrastive focus (in the original, "with contrastive new information"). Lengthening of the stressed syllable was found for all word types in contrastive focus. For Mexico City Spanish, Kim and Avelino (2003) conducted a read-text task with five participants, which looked at differences between broad, narrow, and contrastive focus. For subjects and objects in canonical word order (that is, subject-verb-object, SVO) they found significant durational differences for the stressed syllables of the target words (half of them proparoxytones, half of them paroxytones), but they did not find differences for subjects and objects in non-canonical word order. Specifically, they found that stressed syllables were longer in contrastive focus than in broad focus. From this work, Kim and Avelino conclude that "duration is the most consistent cue in distinguishing contrastive and narrow focus types from broad focus" (p. 372). In this paper, we investigate the issue of post-lexical duration in Yucatecan Spanish in subjects and objects in canonical word order, since there are reasons to doubt that Kim and Avelino (2003) generalization may be extended to the Yucatán peninsula (see below).

One important point to bear in mind in this context is that Yucatecan Spanish has been in close contact with Yucatec Maya for more than 500 years by now. According to the 2010 census of the Mexican National Institute of Indigenous Languages (Instituto Nacional de Lenguas Indígenas-INALI 2010), the population of Mexico speaking Yucatec Maya as their first or second language amounts to approximately 760,000 persons. The great majority of them (758,000 persons) live in the Yucatán peninsula, which includes the federal states of Yucatán, Campeche, and Quintana Roo. According to the last population census of the National Institute of Statistics and Geography (Instituto Nacional de Estadística y Geografía-INEGI 2015) in 2015, the percentage of the Mayan-speaking population living in the Yucatán peninsula amounts to 8% in Campeche, 12% in Quintana Roo, and 27% in Yucatán, that is, an average of 18% across the entire Yucatán peninsula. In 1970, the Mayan-speaking population on the Yucatán peninsula amounted to 55% (Lope Blanch 1987, p. 9f). During the nineteenth and twentieth centuries, contact between Spanish and Mayan increased considerably, as more and more Mayan speakers migrated to urban areas to work as domestic help, vendors, manual laborers, and nannies to Spanish-speaking families (Barnes and Michnowicz 2013; Lipski 2004, p. 99; Michnowicz 2008). According to Lope Blanch (1987), the Yucatán peninsula clearly outranks all other regions in Mexico with respect to both the considerable standing of the indigenous language as well as the rate of bilingual speakers (Lope Blanch 1987, p. 22).

The close contact between these two languages is particularly important for the present purposes due to at least the following three reasons. First of all, Yucatecan Spanish is characterized by a broad range of peculiarities, at all levels of linguistic representation, which are mostly traced back to language contact with Yucatec Maya in the literature. On the phonological level, this is the case for features such as the labialization of final nasals (Michnowicz 2006a, 2006b, 2007, 2008), the aspiration

of voiceless plosives (Lope Blanch 1987; Mediz Bolio 1951; Michnowicz and Carpenter 2013; Suárez Molina 1996), or glottalization (Lope Blanch 1987; Michnowicz and Kagan 2016; Suárez Molina 1996), and it is especially true for the peculiar intonation of the variety (see, e.g., Barrera Vásquez 1977; Mediz Bolio 1951; Michnowicz and Barnes 2013; Suárez Molina 1996):

> [ . . . ] *lo primero que llama la atención del forastero* [ . . . ] *es la entonación fraseal, lenta y pausada, fenómenos que no son sino reflejos de la fonética nativa* [maya]. '[ . . . ] the first thing that attracts the attention of the foreigner [ . . . ] is the unhurried and deliberate pronunciation, which is nothing else than the mirror image of the native [Mayan] phonetics.' (Suárez Molina 1996, p. 63)

Secondly, the issue of language contact deserves further attention in the context of our study of prosodic focus marking in Yucatecan Spanish because Yucatec Maya and (central) Mexican Spanish are typologically unrelated languages that dispose of very different strategies of focus marking. Most importantly, Yucatec Maya is known to have a focus position left-adjacent to the main verb and right-adjacent to an extra-phrasal topic (if present), where focused constituents are generally fronted to (see e.g., Gutiérrez Bravo 2015, pp. 21–23; Kügler and Skopeteas 2007).

3. Tumen to'on-e' [$_{FOC}$ maaya] k t'an-ik-ø
   because 1.PL-TOP Maya HAB.ERG.1PL speak-IND-ABS.3SG
   "Porque nosotros lo que hablamos es MAYA." ["Because what we speak is MAYA"]
   (Gutiérrez Bravo 2015, p. 21; our English translation)

Moreover, Yucatec Maya does not seem to make use of any prosodic means in order to signal the information structural content of syntactic constituents. On the contrary, Gussenhoven and Teeuw (2008), Kügler and Skopeteas (2007), and Kügler et al. (2007) argue that information structure is realized by only syntactic means, that is, fronting to the focus position, in Yucatec Maya. Regarding possible effects of language contact, Gutiérrez-Bravo et al. (2019), building on Sobrino (2010), report on a variety of focus fronting constructions that are entirely ungrammatical in standard Mexican Spanish (cf. (4)).

4. Ellos, VENIR hicieron acá en Yucatán
   they come-INF do.PAST.3PL here in Yucatán
   "They, they came here to Yucatán." (Gutiérrez-Bravo et al. 2019, p. 279)

A detailed comparison of these fronting constructions to their equivalents in Yucatec Maya reveals that the syntax of the Yucatecan Spanish constructions is strikingly similar to the syntax of comparable constructions in Yucatec Maya. Hence, Gutiérrez-Bravo et al. (2019) conclude that sentences such as the one exemplified by (4) are due to direct syntactic transfer from Yucatec Maya. Against this background, we may expect duration (or prosodic cues to focus marking in general) to be much less important in Yucatecan Spanish than in other varieties of Spanish, and since Gutiérrez-Bravo et al. (2019)'s research is based on data from monolingual speakers of Yucatecan Spanish, it suggests that the phenomenon has developed into a regional feature of the variety irrespective of particular language profiles (such as, e.g., monolingual vs. bilingual speakers). This particular issue is further discussed in Section 4 of the present paper.

The third reason for considering language contact in the context of our investigation of post-lexical syllable duration in Yucatecan Spanish is that Spanish and Yucatec Maya furthermore differ with respect to the following three prosodic features. Firstly, Spanish is a language without phonemic vowel length. That is, vowel length is not used to differentiate between word meanings in this language. By contrast, in Yucatec Maya, vowel length is used in order to signal meaning differences at the lexical level (see, e.g., Lehmann 1990; Martínez Huchim 2004; Pike 1946). Table 1 presents some examples of phonemic vowel length in Yucatec Maya.

**Table 1.** Yucatec Maya examples of phonemic short/long vowel contrasts.

| Short Vowel | Long Vowel |
|---|---|
| *chich* [t͡ʃit͡ʃ] 'hard' | *chiich* [t͡ʃiːt͡ʃ] 'granny' |
| *k'an* [k'an] 'yellow' | *k'áan* [k'áːn] 'hammock' |
| *pik* [pik] 'a bug' | *piik* [pìːk] 'petticoat' |

The second crucial differentiating feature is post-lexical duration itself. As mentioned above, there are a considerable amount of studies showing that post-lexical syllable duration is one of the acoustic cues in order to signal post-lexical pragmatic categories in Spanish, such as in particular contrastive focus, whereas Yucatec Maya has been shown to be a language that does not possess any prosodic means of focus marking at all.

Third, duration can be an acoustic cue to word stress. In Spanish, words can be oxytone, paroxytone, or proparoxytone words, that is, stress is contrastive. The main acoustic correlates of stress in this language are F0, duration, and amplitude (see, e.g., Llisterri et al. 2016). To the best of our knowledge, there are no studies on the acoustic correlates of stress in Yucatecan Spanish; however, we estimate that the aforementioned correlates also fit this variety. On the other hand, stress in Yucatec Maya seems to be noncontrastive and it is debated in which position in the word it appears (Frazier 2009, p. 22), although it seems that heavy syllables attract stress (England and Baird 2017). Because of the uncertainty regarding stress in Yucatec Maya and the lack of acoustic studies on the matter (which could shed some light on whether duration is in fact a cue for stress in this language), we cannot make predictions on how stress in Yucatecan Spanish could be influenced by Yucatec Maya.

Importantly for the present purposes, Baird (2017) suggests that phonemic vowel length and (the lack of) post-lexical duration are interrelated in Spanish–Maya contact. He investigated two groups of Spanish–K'ichee' (Maya) bilinguals in Guatemala, one in the community of Cantel and the another one in Nahualá. Cantel K'ichee' has a vocalic system of six vowels that does not possess phonemic vowel length, whereas Nahualá K'ichee', with 10 vowels, does. Baird conducted focus elicitation (production) tasks for both the Spanish and the K'ichee' varieties of the bilingual speakers in said communities. He then measured the duration of the stressed syllables of particular target words in broad focus compared to contrastive focus (oxytones in K'ichee', oxytones and paroxytones in Spanish). As far as the results are concerned, the data obtained for the Mayan varieties suggest, first of all, that duration is used to mark contrastive focus in Cantel K'ichee' but not in Nahualá K'ichee': in other words, the speakers of the former variety employ post-lexical duration to mark contrastive focus, whereas the speakers of the latter do not. Secondly, the results from the Spanish production tasks suggest that the speakers from Nahualá (i.e., those who make use of phonemic vowel length in their Mayan variety) do not use post-lexical duration in order to mark contrastive focus, similarly to what has been observed for their Mayan variety. Contrary to this, the Cantel speakers use post-lexical duration to mark contrastive focus in their Spanish variety just as they do in their variety of K'ichee'. Nevertheless, the analysis revealed differences between female and male speakers of Nahualá in that only female speakers show the lack of post-lexical duration in the Spanish production tasks, and only for oxytone words—in K'ichee', stress appears only in oxytone words (Baird 2014). In view of the fact that these female speakers report using Spanish less than the male ones (according to the Bilingual Language Profile questionnaire that Baird used in this study; see also Section 2.1 below), Baird concludes that the relation between knowledge/proficiency in Spanish and knowledge/proficiency in K'ichee' seems to play a role in the use of post-lexical duration in the Spanish under study. Thus, Maya-dominant speakers of Spanish have phonemic vowel length in their Mayan variety and hence do not make use of post-lexical duration in general (i.e., neither in their Mayan nor in their Spanish languages). On the other hand, Spanish-dominant speakers seem to have lost the ability to phonemically discriminate between different vowel lengths and may thus employ syllable duration as an acoustic cue for post-lexical pragmatic purposes. However, irrespective of the issue of language dominance, Baird (2017) analysis suggests that post-lexical duration as a device of focus

marking in these Spanish varieties is closely related to the presence or absence of phonemic vowel length in the respective K'ichee' Maya contact varieties.

As for Yucatecan Spanish, it is important to recall that it has been in close contact with Yucatec Maya for more than 500 years (see above), and that it presents a similar picture to Nahualá Spanish in that the contact language, Yucatec Maya, has phonemic vowel length but no post-lexical duration to mark focus.

Cross-linguistically, it is not always the case that the presence of phonemic vowel length precludes the use of post-lexical duration to mark focus, since there are also linguistic varieties that have phonemic vowel length but do nevertheless make use of post-lexical duration, such as Serbian; in this variety, stressed syllables are also longer (Smiljanić 2004).[1] Nahualá K'ichee', similarly to Belgrade Serbian, has phonemic vowel length, but does not use duration as a cue to stress (Baird 2014), nor post-lexically, as has already been explained. Thus, Yucatec Maya could be similar to Nahualá K'ichee' in its lack of duration as a cue to stress, which makes it still possible to relate the presence of phonemic vowel length to the lack of post-lexical duration.[2]

A final point to bear in mind is that speaker age and gender have been identified as important variables in various sociolinguistic and dialectal studies on Yucatán Spanish (see, e.g., García Fajardo 1984; Michnowicz 2006a, 2008, 2015; Pfeiler Blaha 1992; Yager 1989, among others), although the results are anything but conclusive when compared to each other. For example, García Fajardo (1984) studied the vowels and consonants of Yucatecan Spanish in Valladolid, Yucatán, with reference to age, gender and sociocultural groups. She studied labial realizations of nasals in absolute final position (e.g., saying "Yucatám" [jukatam] instead of "Yucatán" [jukatan]) and concludes that all speakers utter them, and there is no pattern according to gender or age (pp. 75–76). However, both Yager (1989) and Michnowicz (2008) studied bilabial realizations of nasals in absolute final position in Mérida, Yucatán, and conclude that labialization is more frequent among women than among men. In Yager's study, speakers in their twenties produced more bilabial realizations, whereas in Michnowicz's this can be said about speakers under 50 years of age. In sum, although age and gender seem to play a role, it is unclear what could be hypothesized based on previous studies.

Consequently, against the background of the above insights on language contact, post-lexical duration and phonemic vowel length in Spanish and Maya, the present study addresses the following three research questions by measuring the duration of the target words' stressed syllables: (i) Do speakers of Yucatecan Spanish use syllable duration as a post-lexical acoustic cue to mark contrastive focus? (ii) Does knowledge of Maya (or language dominance) play a role in how speakers differ with respect to their use of post-lexical duration? (iii) Do the variables age and gender have any effect on the use of syllable duration as a post-lexical acoustic cue in order to mark contrastive focus? The hypotheses that can be posited on the basis of the literature reviewed above are: (i) Contrary to what has been observed for Central Mexican Spanish, speakers of Yucatecan Spanish generally do not make use of greater duration of the stressed syllable to mark contrastive focus in their Spanish variety, although inter-individual variation is expected to exist, (ii) in particular, knowledge of Maya might still correlate with a higher degree of lack of syllable duration in order to mark contrastive focus, since the linguistic system of Yucatec Maya provides information with respect to phonemic vowel length that the monolingual speakers of Yucatecan Spanish do not possess (iii) age and gender may have an effect, although it is not possible to deduce any concrete hypotheses from the literature due to the aforementioned reasons.

The outline of the article is as follows. In Sections 2 and 3, we present a production investigation on Yucatecan Spanish (Section 2) as well as the results (Section 3). In Section 4, we discuss the results

---

[1] We thank a reviewer for this useful piece of information.
[2] Secondly, we might draw our attention to other contact varieties of Spanish, since it has been found that Peruvian Spanish does not seem to have post-lexical duration either (Muntendam and Torreira 2016). However, this issue is beyond the scope of the present paper so we have decided to tackle it in future research.

against the background of the presumed interrelation of phonemic vowel length and lack of post-lexical duration presented by Baird (2017) and we mention several points worthy of further study in future research. Finally, in Section 5, we present the main conclusions.

## 2. Materials and Methods

### 2.1. Participants

This study focuses on Yucatecan Spanish as it is spoken in the state of Quintana Roo, specifically in the municipality of Felipe Carrillo Puerto. In Felipe Carrillo Puerto, 21.2% of the population speak Yucatec Maya (Gobierno del Estado de Quintana Roo-Secretaría de Desarrollo Social 2010). Forty-one speakers of Yucatecan Spanish were recorded for this study. All subjects gave their informed consent for inclusion before they participated in the study. Seven were excluded, which resulted in a total of 34 participants (18 females and 16 males, age range = 17–84). Speakers were excluded either because they did not understand the task properly, they hesitated throughout the task, or they answered most questions in a way that would not allow for a comparison with the other speakers' utterances (for example, by describing the slide at length instead of giving a concrete answer to it). As will be indicated in Section 2.2, literacy was necessary to complete the task; however, because all participants were literate in Spanish, none were excluded.

Participants' language dominance was assessed by means of the Bilingual Language Profile, or BLP (Birdsong et al. 2012). The BLP is a thorough questionnaire made up of questions about language history, use, proficiency, and attitudes. It yields a score quantifying language dominance; each end of the score continuum (−/+218) corresponds to the highest dominance for a given language—in the present study, Maya and Spanish, respectively—whereas values around 0 indicate balanced bilingualism. Figure 1 shows the scores corresponding to each participant. The scores are distributed along the continuum; values closer to the highest Mayan dominance (e.g., −200, −150) would represent speakers who speak little or no Spanish, which explains the absence of such values.

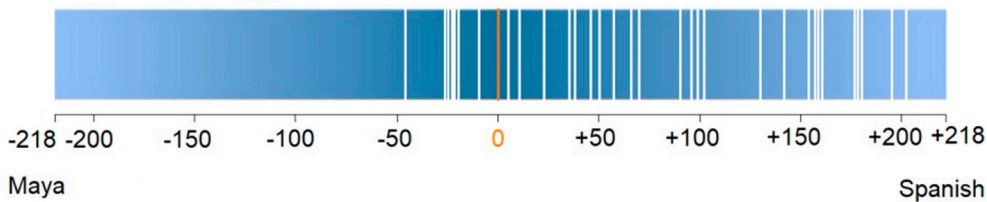

**Figure 1.** Bilingual Language Profile scores of participants (N = 34). Each individual score is indicated by a white vertical line.

None of the 34 speakers fall into the −50–−150 range, which would have been desirable considering that there is clearly variation in BLP scores on the Spanish dominant side. It must be kept in mind that BLP scores were obtained in the same session as the recordings, that is: BLP scores were not used to decide which participants should be included or excluded from the study. Although a number of participants was recruited under the assumption that they spoke both languages to different degrees, the actual BLP scores were computed only after the recordings took place. Appendix A lists all BLP scores, gender, age of participants, and number of observations (both for subject and object positions).

### 2.2. Experimental Design

The duration of the target words' stressed syllables in both broad and contrastive foci, both in subject and object positions in SVO-sentences, constitutes the focus of this study. Words in subject and object position were chosen to obtain data from two positions in the utterance, namely initial and final. If greater duration is used to mark contrastive focus, the results should be comparable in the two positions. For example, in his study of Madrid Spanish, Face (2002) found differences in the duration of the stressed syllable between broad and contrastive focus in all utterance positions considered in his

study (i.e., initial, medial, and final). Consequently, we would expect that utterance-final lengthening does not play a role in the results of words in object position. In addition, we predict comparable results for subjects (in utterance-initial position) and objects (in utterance-final position). Moreover, although a certain degree of variation was expected, it was also expected that the most frequent word order would be SVO, meaning that subjects appear in utterance-initial position while objects appear at the end of the utterance.

Participants were shown a slide presentation intended to elicit broad or contrastive foci responses for words in subject and object positions. First, participants were shown an example of a slide, similar to that in Figure 2. Then, they were given several suggestions about possible ways to answer. They were instructed to give only one answer, which should be a whole sentence, and to answer it in a natural way; they were also told that there were no incorrect answers. Second, they were shown two trial examples (similar to that of Figure 2), one intended to elicit (subject or object) contrastive focus and the other one, broad focus. Finally, they proceeded to answer to the questions of the task proper.

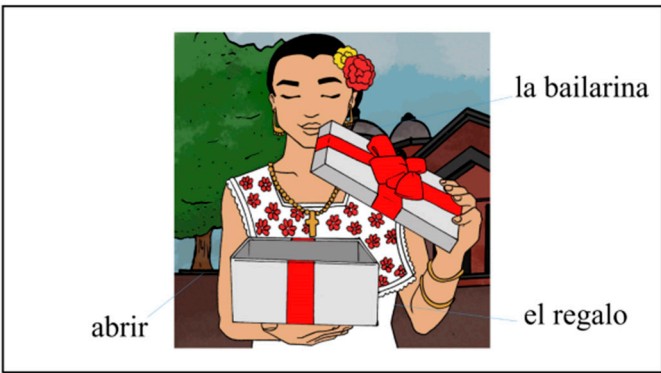

*El abogado está abriendo el regalo, ¿verdad?*

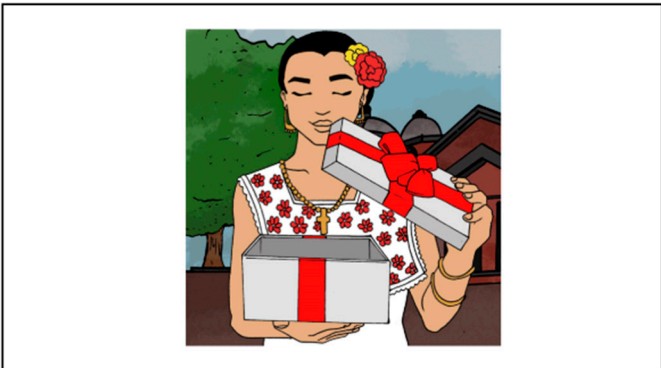

**Figure 2.** Example from the production task. Above: slide with lexical items. Below: slide plus researcher's question to elicit contrastive focus on the subject (Translation: 'The lawyer is opening the present, right?').

Figure 2 is an example of a pair of slides that the participants were shown. In this example, contrastive focus for the word in subject position is elicited. The procedure was as follows: first, a participant would see an image (Figure 2) where the three lexical items corresponding to the action taking place would appear. Thus, a verb in infinitive form plus two names preceded by the corresponding definite article (one for the subject and the other for the object) appeared on this slide so that the participant would use them. It was made sure that the order and position in which the three lexical items were shown on the slide varied as much as possible across slides so as to avoid a reading style. This reading style could have appeared if the words had always been written in the same order (i.e., from top to bottom, the words for the subject, the verb, and the object). Providing the words in written form was considered a necessary compromise between spontaneity and the elicitation of the same lexical items from all participants. In the next step, one of the researchers asked the participant a

question related to the image provided, this time without any written text. Second, the question in Figure 2 (below) aimed at eliciting contrastive focus for the subject: *El abogado está abriendo el regalo, ¿verdad?* "The lawyer is opening the present, right?" One possible answer to this question would be: *(No,) la bailarina está abriendo el regalo* "No, the dancer is opening the present." Slides intended to elicit broad focus had the same structure, but the question used was always the same (*¿Qué pasa aquí?* "What's happening here?").

Broad and contrastive questions were presented in a pseudo-randomized order to control for habituation effects and also to avoid a reading style. Target words for subjects (*abogado* 'lawyer', *bailarina* 'dancer', *enfermera* 'nurse', *marinero* 'sailor', *vendedora* 'seller'; stressed syllables are indicated in boldface) referred to occupations; objects (*ballena* 'whale', *bombero* 'firefighter', *granada* 'pomegranate', *merengue* 'meringue') were of a more varied nature. In the broad focus context, target words included both subjects and objects; also, more target words were used for the subject condition, which accounts for the fact that there are more instances of subject target words in the broad focus than in the contrastive focus condition. There was another subject target word (*panadero* 'baker') which was not taken into account in the present study because examples of it appeared only in one of the focus conditions. Similarly, other words were also used for objects, but were excluded because they appeared in only one focus condition. In sum, 27 questions that elicited subject target words (17 for broad focus, 10 for contrastive focus) and 8 that elicited object target words (4 for broad focus, 4 for contrastive focus) were taken into account in this investigation.

Crucially, what was measured were not the target words, but the stressed syllables of said target words (e.g., "*ri*" in *bailarina*). *Target word* is used in the article for ease of explanation. All target words were trisyllabic, paroxytone words (the paroxytone stress pattern is the most common in Spanish); thus, the position of the target syllables in the word, and ultimately in the utterance, was further controlled. Although the segmental make-up of the studied syllables differs, it allows for an analysis of pitch characteristics. This possibility is not undertaken in this study.

Recordings (44.1 kHz, 16 bit, wav) were made in a silent room with an AKG C 544 L head-mounted microphone connected to a Presonus Audiobox USB in the presence of at least one of the researchers.

*2.3. Data Analysis*

Recordings were orthographically transcribed by a native speaker of Yucatecan Spanish. They were then automatically aligned using BAS Pipeline online service (v. 2.22; Kisler et al. 2017). Later, they were manually corrected by two labelers in Praat (v. 6.0.43; Boersma and Weenink 2009) under the supervision of the first author, who also conducted final corrections on the labels to ensure accuracy and coherence in the measurement. Target syllables were labeled based on visual inspection of the spectrograms and waveforms; the boundaries were put at the nearest zero crossing on the waveform. For the extraction of the duration, a script (Hirst 2009) was used. Figure 3 shows an example of the labeling performed.

Of the possible total of 918 answers for target syllables in subject position and 272 in object position, 54 (5.9 %) and 10 (3.7 %) were excluded, respectively. Syllables were excluded because (i) a different word instead of the target word was uttered (e.g., *señor* 'man' instead of *abogado* 'lawyer'; 4 in utterances where the target word appeared in subject position, 0 in object position); (ii) participants hesitated in a way that could influence the duration of the target word (16 in subject position, 1 in object position); or (iii) participants reversed the roles of subject and object (e.g., *El bombero está llamando a la bailarina* 'The firefighter is calling the dancer' instead of *La bailarina está llamando al bombero* 'The dancer is calling the firefighter'; 10 in subject position, 7 in object position). In addition, several utterances were excluded for syntactic reasons, specifically because of clefting (e.g., *No, es la vendedora la que vende el helado* 'No, it is the seller who is selling the ice cream'; 5 in subject position, 0 in object position), fronting (e.g., *No, la vela la está vendiendo la vendedora* 'No, the candle is being sold by the seller'; 12 in subject position, 1 in object position), or because they used a VOS word order (*Lava la herida la enfermera* 'Washes the wound the nurse'; 4 in subject position, 1 in object position).

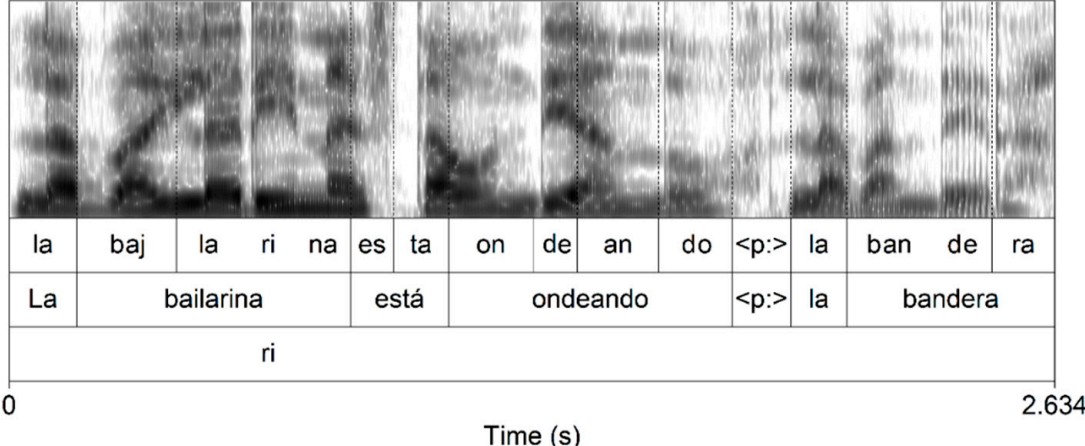

**Figure 3.** Example of labeling of the target syllable (**ri**) in the broad focus utterance *La bailarina está ondeando la bandera* 'The dancer is waiving the flag', uttered by a female speaker.

Sentences where the indefinite article was used (e.g., *Una enfermera está lavando una herida* 'A nurse is washing a wound') were also kept. The indefinite and definite articles in Spanish differ in that the first ones are lexically stressed (Hualde 2009; Quilis 1999). In principle, this difference should not affect the target syllable. Nevertheless, previous studies have reported that the indefinite article can receive accentual prominence instead of the word it complements (Muntendam and Torreira 2016). Consequently, all contrastive focus sentences with indefinite articles were further checked auditorily by the first author, a native speaker of (European) Spanish, to make sure that no such cases would be included in the analysis. As a result, 3 target syllables (all uttered by the same speaker in subject position) were excluded. In sum, the analysis was conducted on a total of 864 stressed syllables in subject position and on 262 in object position.

## 3. Results

We measured the duration of the stressed syllables of the 1126 target words in subject position (864 syllables) and object position (262 syllables) in SVO sentences in broad and contrastive focus sentences uttered by 34 speakers of Yucatecan Spanish with different degrees of Maya–Spanish dominance according to BLP scores (Birdsong et al. 2012; see Section 2.1). The measurements were submitted to two separate generalized linear mixed-effects models (one for subject position, another for object position) using the `lmer` function in the `lme4` package (Bates et al. 2015) in R (version 3.5.1; R Core Team 2018), with syllable duration (in ms) as the (continuous) dependent variable, and with the following independent variables: focus condition (categorical: broad vs. contrastive), BLP score (continuous), gender (categorical: male vs. female), and age (interval). Random intercepts and random slopes for focus were fitted both for speakers and target syllables. Satterthwaite approximations to degrees of freedom were used to calculate *p* values. We also checked for two-way interactions between BLP score and focus by means of a likelihood ratio test that compared the model that included the interaction with the model without the interaction.

### 3.1. Results for Subject Position

Table 2 shows the mean duration and standard deviation for each stressed syllable in subject position.

**Table 2.** Descriptive results for the stressed syllables (in boldface) of the target words in subject position. Mean duration and standard deviations (SD) are indicated in ms.

| Target Syllable | *n* | Mean (SD) |
|---|---|---|
| abo**ga**do | | |
| broad focus | 32 | 182.51 (59.6) |
| contrastive focus | 29 | 195.1 (49.78) |
| baila**ri**na | | |
| broad focus | 129 | 131.82 (30.36) |
| contrastive focus | 89 | 135.54 (33.5) |
| enfer**me**ra | | |
| broad focus | 126 | 177.04 (49.25) |
| contrastive focus | 65 | 194.32 (42.9) |
| mari**ne**ro | | |
| broad focus | 98 | 191.38 (47.6) |
| contrastive focus | 62 | 206.43 (34.32) |
| vende**do**ra | | |
| broad focus | 171 | 182.96 (47.03) |
| contrastive focus | 63 | 179.69 (38.65) |

The model for subject position shows that there is no effect of focus nor of BLP score (Table 3). Older speakers seem to utter longer target syllables. There is no interaction between BLP score and focus (likelihood ratio test: $p = 0.44$). Figure 4 provides information about the duration of each stressed syllable in broad and contrastive focus conditions. Figure 5 shows the duration of said syllables in broad and contrastive focus conditions per speaker. Instances of stressed syllables are represented by circles, blue for broad focus condition and orange for contrastive focus condition. Each vertical line represents a speaker. "Blue" and "orange" circles do not show any pattern for any of the speakers. The horizontal line represents the mean duration value for stressed syllables in broad focus condition (in blue) and contrastive focus condition (in orange) as a function of BLP score. Both the representation for each speaker and the mean duration values show the lack of interaction between BLP score and focus.

**Table 3.** Results of the mixed-effects model for stressed syllables in subject position ($N = 864$).

| Fixed Effect | Estimate | SE | *t* | *p* |
|---|---|---|---|---|
| (Intercept) | 154.77 | 18.55 | 8.34 | <0.001 |
| focus | 6.62 | 5.20 | 1.27 | 0.23 (n.s.) |
| BLP score | −0.08 | 0.07 | −1.06 | 0.30 (n.s.) |
| gender | −3.10 | 9.86 | −0.32 | 0.76 (n.s.) |
| age | 0.65 | 0.30 | 2.14 | 0.04 |
| focus BLP score | 0.02 | 0.04 | 0.60 | 0.55 (n.s.) |

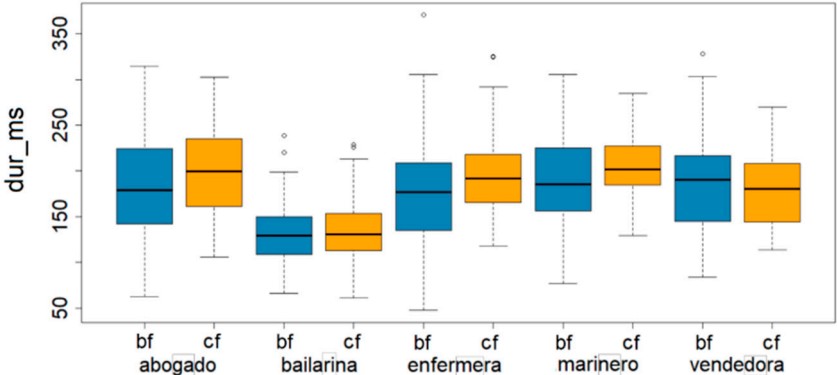

**Figure 4.** Duration (in ms) of the stressed syllables of the target words in subject position in broad focus ("bf") and contrastive focus ("cf") conditions.

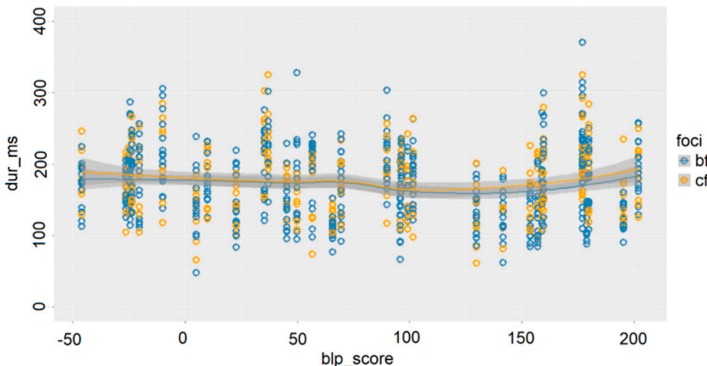

**Figure 5.** Duration (in ms) of the stressed syllables of the target words in subject position in broad focus ("bf") and contrastive focus ("cf") conditions in relation to Bilingual Language Profile (BLP) scores. Each vertical line corresponds to a participant's BLP score.

*3.2. Results for Object Position*

Table 4 shows the mean duration and standard deviation for each stressed syllable in object position.

**Table 4.** Descriptive results for the stressed syllables (in boldface) of the target words in object position. Mean duration and standard deviations (SD) are indicated in ms.

| Target Syllable | *n* | Mean (SD) |
|---|---|---|
| ba**lle**na | | |
| broad focus | 32 | 181.52 (44.61) |
| contrastive focus | 33 | 197.36 (37.95) |
| bom**be**ro | | |
| broad focus | 31 | 114.22 (26.62) |
| contrastive focus | 31 | 121.12 (25.68) |
| gra**na**da | | |
| broad focus | 34 | 190.3 (37.56) |
| contrastive focus | 33 | 185.25 (37.65) |
| me**ren**gue | | |
| broad focus | 34 | 222.81 (34.29) |
| contrastive focus | 34 | 228.9 (40.28) |

The model for object position shows that there is no effect of focus nor of BLP score. There is a difference between female and male speakers, with male speakers producing shorter target syllables. There is no interaction between BLP score and focus (likelihood ratio test: $p = 0.52$). Figure 6 provides information about the duration of each stressed syllable in broad and contrastive focus conditions. Figure 7 shows the duration of said syllables in broad and contrastive focus conditions per speaker.

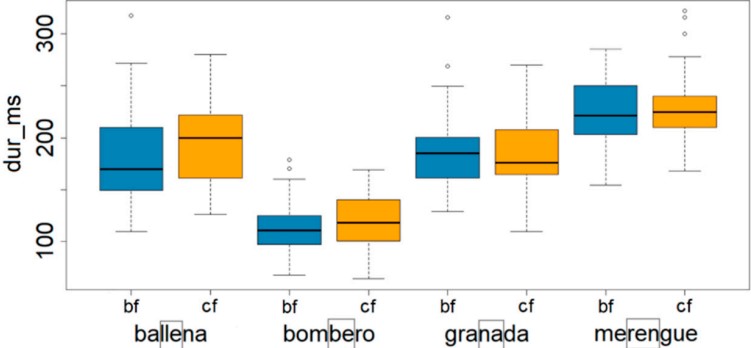

**Figure 6.** Duration (in ms) of the stressed syllables of the target words in object position in broad focus ("bf") and contrastive focus ("cf") conditions.

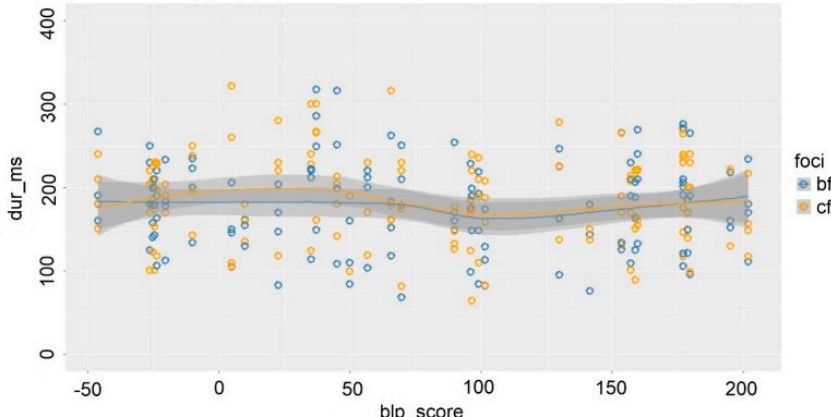

**Figure 7.** Duration (in ms) of the stressed syllables of the target words in object position in broad focus ("bf") and contrastive focus ("cf") conditions in relation to Bilingual Language Profile (BLP) scores. Each vertical line corresponds to a participant's BLP score.

*3.3. Summary of Results*

Tables 2 and 4 and Figures 4–7 show that the mean duration of almost all stressed syllables is longer in contrastive than in broad focus. Nevertheless, they also show large standard deviations and great inter-speaker variation. The mixed-effects models for the duration of stressed syllables in subject (utterance-initial) and object (utterance-final) position yield the same result: speakers of Yucatecan Spanish do not employ post-lexical duration (Tables 3 and 5).

**Table 5.** Results of the mixed-effects model for stressed syllables in object position ($N = 262$).

| Fixed Effect | Estimate | SE | t | p |
|---|---|---|---|---|
| (Intercept) | 174.27 | 25.52 | 6.83 | 0.02 |
| focus | 8.60 | 6.31 | 1.36 | 0.21 (n.s.) |
| BLP score | −0.01 | 0.06 | −0.22 | 0.82 (n.s.) |
| gender | −22.00 | 7.01 | −3.14 | 0.004 |
| age | 0.34 | 0.22 | 1.60 | 0.12 (n.s.) |
| focus BLP score | −0.03 | 0.05 | −0.61 | 0.54 (n.s.) |

In summary, the results in the previous subsections provide the following answers to the three hypotheses of the study (see Section 1): (i) Speakers of Yucatecan Spanish do not use duration (of the stressed syllable) as a post-lexical acoustic cue in order to mark contrastive focus, neither in subject nor in object position; (ii) knowledge of Maya does not seem to play a direct role in how speakers differ with respect to their use of post-lexical duration, neither in subject nor in object position; finally, (iii) some differences are observed in regard to age (older speakers utter longer syllables in subject position, but not in object position) and gender (male speakers utter shorter target syllables in object position, but not in subject position). In sum, hypothesis (i) is supported by the results, whereas hypothesis (ii) is not. Hypothesis (iii) is only partially supported by the results, with age and gender having some effect on post-lexical duration.

**4. Discussion**

Keeping in mind Baird (2017) argumentation about Nahualá K'ichee' and Gutiérrez-Bravo et al. (2019) work on the syntactic transfer of Yucatec Mayan focus fronting constructions into Yucatecan Spanish, the obvious question is if the lack of post-lexical duration in Yucatecan Spanish may be related to cross-linguistic influence from Yucatec Maya, and, if yes, how exactly this influence may have originated across the years. On the one hand, the situation of Yucatecan Spanish is analogous to the Nahualá case in that in both contact situations there is an indigenous contact language with phonemic

vowel length and a Spanish contact variety that does not show any effects of post-lexical duration. However, contrary to what Baird (2017) reports for the female Nahualá speakers, the bilingual speakers of our experiment were balanced bilinguals, and as such, they were as highly proficient in Spanish as they were in Yucatec Maya, meaning that no effect could be attributed to a reduced knowledge of Spanish to begin with.

Moreover, the lack of a direct effect of the Bilingual Language Profile (BLP) scores on the duration variable shows that the phenomenon at stake here cannot be attributed to any particular degree of Maya proficiency of the current speakers either. However, it is important to note that the region at stake here was entirely Maya-dominated until the end of the Caste War in 1901.[2] Indigenous uprisings continued on a smaller scale for several decades (Grube 2000, p. 420; Reed 1968), and the supra-regional activities in the regional commercial sector were considerably curtailed by the resistance of large parts of the Mayan population (see, e.g., Bracamonte y Sosa 1986), the socialist movement, and land reforms by Salvador Alvarado (1915–1918) and Felipe Carrillo Puerto (1922–1924) (Higuera Bonfil 1986; Joseph 2003; Scheuzger 2009, among others). Most importantly for the present purposes, Spanish schools were not implemented in the region until the 1930s for various reasons (Eiss 2004; Fallaw 2004; Hostettler 1999). Consequently, the institution-driven hispanization of the area only started around 1940–1950, meaning that, although they have mastered Spanish at a native or near-native level, the majority of the current population of Felipe Carrillo Puerto acquired Spanish either as a second language (L2) or as a first language (L1) from L2 speakers or immediate descendants of L2 speakers. Experts in language contact studies such as Winford (2014), Thomason (2001), and Van Coetsem (1988), among many others, assume that this type of language shift scenario favours phenomena related to "second-language acquisition strategies" (Thomason 2001, p. 60), including interlanguage phenomena and transfer (Thomason 2001, p. 60), due to the fact that the strategies emanate from the speakers of the source language. From this perspective, and in view of the particularity of the lack of post-lexical syllable duration in this variety compared to other varieties of Spanish (see Section 1), it is still probable that the particular nature of the syntax–prosody interface of the corresponding Yucatec Maya speakers did indeed have an impact on the prosodic realization of contrastive focus in the Spanish variety that is spoken by the current Spanish speakers of that region. Since the Spanish of the region has been influenced, and shaped, by three generations of Maya speakers, at least the first generation of which was definitely dominant in Yucatec Maya, the lack of post-lexical duration may very well have originated as a second language interference phenomenon and may then have passed on to the native Spanish of the subsequent generations during their acquisition of Spanish as a first language, so that it developed into an integral part of what is nowadays called Yucatecan Spanish due to cross-linguistic influence (without there being any *current* influence of the contact language on post-lexical syllable duration in Yucatecan Spanish).

Another intricate issue concerns the relation between the lack of post-lexical duration and the apparent importance of syntactic means of focalization in Yucatecan Spanish suggested by the syntactic investigation of Gutiérrez-Bravo et al. (2019). In view of the patterns reported in this latter work, it might be questioned whether the lack of post-lexical duration in our experiment is to be attributed to a cross-linguistic phonological influence as described above, or if it is rather due to the high(er) importance of syntactic focus marking, thus being an indirect effect of another contact-related particularity. However, note, first of all, that the speakers in our experiment did not show any marked preference for syntactic strategies of focus marking (a fact that is probably due to the rather controlled, and hence formal, nature of the experimental setting, although this did not hinder the use of some non-SVO sentences by some participants). Moreover, the intricate relation between the lack of post-lexical duration and the preference for syntactic over prosodic focus marking does obviously

---

[2]  Secondly, we might draw our attention to other contact varieties of Spanish, since it has been found that Peruvian Spanish does not seem to have post-lexical duration either (Muntendam and Torreira 2016). However, this issue is beyond the scope of the present paper so we have decided to tackle it in future research.

hold for both Yucatecan Spanish and Yucatec Maya alike, both features being generally considered two sides of the same coin in the works on focus realization in Yucatec Maya (see Gussenhoven and Teeuw 2008; Kügler et al. 2007, among others). Thus, it may be argued that this interrelation extended from Yucatec Maya to Yucatecan Spanish together with the two isolate features or tendencies, respectively, meaning that all three of them are best to be considered as cross-linguistic particularities of the entire region, comparable to those non-genealogically determined similarities between neighboring languages that have been uncovered in the tradition of areal linguistics for various linguistic areas (see, e.g., (Bisang 2006; Campbell 2006; Hickey 2017), and the references cited therein).

It is true that, since our study leads to a negative result (i.e., that speakers of Yucatecan Spanish do not seem to use post-lexical duration), it is notably confronted with the issue of validity. That is, it might be objected that the speakers simply failed to show effects of post-lexical duration, since they did not properly re-enact the intended pragmatic condition/s (particularly in the context of correction and contrastivity). Although this problem may never be entirely excluded in semi-spontaneous production tasks, we have reasons not to be too pessimistic in this regard. A crucial piece of evidence has already been alluded to in Section 2.3, where we saw that several utterances were disregarded because they did not comply with the SVO order. The largest group of utterances was that of fronted constituents (12 where the target word was in subject position, 1 in object position), followed by cleft constructions (5 in subject position). The fact that other ways of marking focus have been used in the corpus is an index of the degree of spontaneity obtained in spite of the somehow experimental task. This provides further validity to the results of the study. Thus, we are inclined to think that the study does indeed provide valid evidence against the use of post-lexical duration in Yucatecan Spanish.

The Bilingual Language Profile (BLP) scores of the participants (Figure 1) show that bilingual speakers' knowledge of Maya and Spanish was arguably balanced. As indicated in Section 2.1, BLP scores were computed after the recordings were made; also, literacy in Spanish was required to complete the task. Consequently, people who could speak both Maya and Spanish but that were not literate in the second could not be included. It is unclear whether having such participants in the study would have had an effect on the results. Although we cannot exclude this possibility, the number of participants (seven) with a "—score" (i.e., more dominant in Maya) may be large enough to sustain our interpretation of the results.

Age and gender deserve further discussion. For the duration of syllables in subject position, older speakers seem to utter longer target syllables. This could be easily explained if we were to pose that older speakers have a slower speech rate; however, this interpretation is contradicted by the results for object position, for which no such tendency is observed. In addition, in object position, male speakers seem to produce shorter target syllables than female speakers; however, this is not paralleled in the subject group, nor does there seem to be an immediate explanation for it. Baird (2017) also found different results for female and male speakers (see Section 1), but he related it to female speakers having a better knowledge of K'ichee'; this situation is not the same as ours.

Our study takes into account the duration of stressed syllables of paroxytone words in subject and object position exclusively. Breen et al. (2010) consider that studying only one acoustic feature of single words is an important limitation in the study of the relationship between prosody and information structure because it leaves out the context in which the target words appear (p. 1052). In this line of argumentation, it could be considered that duration—and also other acoustic cues—should be measured not only for the stressed syllable, but also for the word it appears in and even in a broader context such as the prosodic constituent. However, the work by De la Mota Gorriz (1997) already reviewed in Section 1 did not only take into consideration the whole utterance, but also several F0 measures, and still found differences in duration between contrastive and non-contrastive (broad) focus. Nevertheless, in a multidimensional view it is the conjunction of several cues which may indicate differences in focus marking. Even if duration by itself is not enough to distinguish between broad and contrastive focus, be it absolute (such as in this study) or relative (see, e.g., Muntendam and Torreira 2016, or Van Rijswijk et al. 2017) it may do so if duration is taken into consideration with other

acoustic cues such as intensity, F0 height or alignment, and also by taking into account the context. This possibility could be addressed in further research.

Finally, a complementary possibility is the insight that a perceptual study might provide. If indeed there are prosodic differences between broad and contrastive focus, even if they are not durational ones, or differences from the conjunction of duration and other acoustic cues, they may be apparent to Yucatecan Spanish listeners. Further research may shed some light on this.

## 5. Conclusions

In this article, we dealt with the duration of stressed syllables in broad versus contrastive focus in Yucatecan Spanish and discussed its connection with Spanish–Maya bilingualism. We studied the duration of stressed syllables of nouns in subject and object position in 1126 SVO sentences (broad and contrastive focus) of a semi-spontaneous production task run with 34 mono- and bilingual speakers in Quintana Roo, Mexico. The main result of the study is that, although slightly longer in contrastive focus, syllable duration turned out not to be a significant variable in our data set, suggesting that duration does not constitute a cue to focus marking in Yucatecan Spanish. Finally, it was discussed how this result relates to the strong influence of Yucatec Maya on Yucatecan Spanish prosody observed by both scholars and native speakers of Yucatecan Spanish and other Mexican varieties of Spanish. In particular, the discussion concentrated on the claim that phonemic vowel length in a language might prevent the use of syllable duration as a post-lexical acoustic cue in another, as suggested by Baird (2017) for the variety of Nahualá Spanish spoken by the Maya-dominant women of the corresponding Guatemalan community. However, since the arguments in favor of such an interdependence are less conclusive in the case of Yucatecan Spanish than in the Nahualá case, we argued that the most evident conclusion that may be drawn from our analysis would rather be that the lack of post-lexical duration in Yucatecan Spanish is a regional feature that (i) is characteristic of both Yucatec Maya and Yucatecan Spanish alike, and that (ii) originated as a second language interference phenomenon and then passed on to the native (Yucatecan) Spanish of the subsequent generations during their acquisition of Spanish as a first language. Interestingly, Yucatecan Spanish has been argued to be similar to Yucatec Maya in that (syntactic) focus fronting is rather high and seems to show Mayan influence in this variety. From this perspective, it might be interesting to pursue the hypothesis that the cross-linguistic influence extends to the entire prosody–syntax interface, and that the corresponding interrelated features are cross-linguistic particularities non-genealogically determined of the entire region. To us, this seems an interesting hypothesis to be pursued in future research.

**Author Contributions:** Conceptualization, N.M.G. and M.U.; Data curation, N.M.G.; Formal analysis, N.M.G. and M.U.; Funding acquisition, M.U.; Investigation, N.M.G. and M.U.; Methodology, N.M.G. and M.U.; Project administration, M.U.; Supervision, M.U.; Visualization, N.M.G.; Writing—original draft, N.M.G. and M.U.; Writing—review & editing, N.M.G. and M.U.

**Funding:** This research is funded by the *Deutsche Forschungsgemeinschaft (DFG)* as part of the CRC 1252 "Prominence in Language", project A5 "Prominence marking and language contact in Spanish", directed by Melanie Uth.

**Acknowledgments:** This paper was presented at the *Bilingualism in the Hispanic and Lusophone World* conference, held in Leiden University on 9–11 January 2019. The authors are most grateful to the audience of the conference and to the two reviewers of its abstract for their feedback. We also wish to thank the two anonymous reviewers of the present article for their highly valuable comments and suggestions. Finally, we thank Patrick Auhagen (for the annotation and correction of the recordings, as well as for the modification of the task pictures), Leonard Rick (for the annotation of the recordings), and Anna Wördehoff (for the correction of the transcriptions).

**Conflicts of Interest:** The authors declare no conflict of interest.

# Appendix A

**Table A1.** Participants' Bilingual Language Profile scores (BLP scores), gender, age, and number of observations (both for subject and object positions). BLP scores quantify language dominance; each end of the score corresponds to the highest dominance for Maya (−218) and Spanish (+218), respectively. Values around 0 indicate balanced bilingualism.

| | | | | Number of Observations | | | |
| | | | | Subject | | Object | |
| Participant ID | BLP Score | Gender | Age | Broad Focus | Contrastive Focus | Broad Focus | Contrastive Focus |
|---|---|---|---|---|---|---|---|
| 01 | 95.994 | male | 22 | 17 | 10 | 4 | 4 |
| 02 | 153.742 | male | 18 | 16 | 10 | 4 | 4 |
| 03 | 129.86 | female | 20 | 17 | 10 | 4 | 4 |
| 04 | 65.754 | female | 31 | 17 | 10 | 4 | 4 |
| 05 | 101.618 | male | 21 | 18 | 9 | 4 | 4 |
| 06 | 37.146 | female | 84 | 16 | 9 | 4 | 4 |
| 07 | 202.05 | female | 54 | 17 | 9 | 4 | 4 |
| 08 | 96.444 | male | 72 | 13 | 10 | 3 | 4 |
| 10 | 158.92 | male | 38 | 16 | 10 | 4 | 4 |
| 11 | 195.24 | female | 67 | 16 | 10 | 2 | 2 |
| 12 | 178.896 | male | 34 | 14 | 2 | 4 | 4 |
| 13 | 177.08 | female | 36 | 17 | 9 | 4 | 4 |
| 14 | 9.992 | male | 46 | 16 | 9 | 3 | 3 |
| 15 | −26.246 | female | 43 | 16 | 9 | 4 | 4 |
| 17 | −45.952 | female | 30 | 16 | 10 | 4 | 4 |
| 18 | −23.702 | female | 38 | 17 | 10 | 4 | 4 |
| 19 | −24.426 | female | 34 | 17 | 6 | 4 | 4 |
| 20 | −20.434 | male | 47 | 15 | 9 | 4 | 3 |
| 22 | −25.146 | male | 38 | 17 | 7 | 4 | 4 |
| 23 | 99.172 | female | 37 | 16 | 10 | 4 | 4 |
| 24 | 141.668 | male | 47 | 16 | 7 | 3 | 3 |
| 25 | 35.334 | female | 44 | 17 | 9 | 4 | 4 |
| 26 | 56.764 | female | 41 | 17 | 10 | 4 | 4 |
| 27 | 22.798 | female | 58 | 17 | 10 | 4 | 4 |
| 28 | 177.08 | female | 63 | 17 | 10 | 4 | 4 |
| 29 | 159.828 | male | 66 | 15 | 10 | 4 | 4 |
| 30 | 157.104 | male | 46 | 17 | 10 | 4 | 4 |
| 31 | 179.804 | female | 34 | 17 | 10 | 4 | 4 |
| 32 | 49.858 | male | 33 | 16 | 9 | 4 | 4 |
| 36 | 69.658 | female | 26 | 17 | 6 | 4 | 4 |
| 38 | 89.906 | male | 17 | 17 | 10 | 4 | 4 |
| 39 | 4.99 | male | 20 | 15 | 10 | 4 | 4 |
| 40 | −9.984 | male | 18 | 17 | 10 | 4 | 4 |
| 41 | 45.228 | female | 43 | 17 | 9 | 4 | 4 |

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
