# Peer review of "Lack of Syllable Duration as a Post-Lexical Acoustic Cue in Spanish in Contact with Maya"

_languages, doi:10.3390/languages4040084_

Round 1

Reviewer 1 Report

Overall, I think the paper is sound and interesting. The conclusions are on point with the data presented. However, I suggest the following before publication.

1. The writing needs work and is unclear in some places.

2. Table 1: words should be written in IPA. A double vowel is sometimes used in Mayan orthography to denote a long vowel; however, several Mayan languages can have a sequence of two identical vowels. IPA will clear this up.

3. The introduction section is missing citations to key studies on the prosody of Yucatecan Spanish. The authors need to consult works of scholars such as Jim Michnowicz and colleagues, whose findings would be beneficial to the explanation of the findings presented here. For example, several of these studies show results that parallel the findings among male speakers shown here.

4. P. 3. Line 100-1,: “knowledge of K’ichee’ does indeed seem to play a role”. Wouldn’t it be a lack of knowledge in Spanish? Or less dominant in Spanish? All of the participants were bilingual, thus all have knowledge of both languages. Also lines 377-9.  The claim in Baird is that the Nahualá females have less knowledge of Spanish, not more of K’ichee’.

5. The Baird (2017) study also finds that this “transfer” primarily occurs in oxytone Spanish words and the argument made is that since stress is fixed in word-final position in K’ichee’, transfer is more likely here. The author’s should justify the use of paroxytone and proparoxytone words in this study. Although stress is Yucatec predictably occurs on heavy syllables (England & Baird 2017), how does this affect Yucatecan Spanish? It is a more complex claim than Spanish-K’ichee’, but needs to be addressed.

6. The Smiljanić (2004) study analyzes two dialects of Serbian: one with phonemic vowel length and one without. Indeed both dialects, regardless of vowel length, mark focus with a longer duration. However, in contrast to Baird (2017), both of these Serbian dialects also mark tonic syllables with a longer duration. The same is not true in K’ichee’, where tonic syllables are not marked by a longer duration in Nahualá but are marked by a longer duration in Cantel (Baird 2014). This appears to be a common feature of Mayan languages. For example, Berinstein (1979) found that Kaqchikel, which has lost Proto-Mayan vowel length, marks tonic syllables with a longer duration whereas Q’anjob’al, which maintains vowel length, does not employ a longer duration to mark tonic syllables. Thus, it may be more than just the presence, or lack thereof, of phonemic vowel length in the contact language, but how duration is used in other linguistic aspects, such as marking stress. The acoustic of stress in Yucatec are still controversial, but several claim that duration is not involved (England & Baird 2017).

7. Methods- the use of a reading task is quite common in these types of studies. However, it does impede the participation of more individuals, particularly those how may not be literate yet still be important factors in the phenomenon under study. I have no issue with the task, but the authors need to acknowledge the shortcomings of it. For example, the majority of the participants in this study are Spanish-dominant according to the BLP and Figure 1. It may be that by using a reading task, more Yucatec-dominant bilinguals were excluded, as literacy and education are often correlated with Spanish use in these areas. This may be a factor in why BLP was not a significant factor in the analysis as most speakers fell on the same side of the dominance continuum.

8. P. 10 lines 374-6. No length differences are found between younger and older speakers for object. However, wasn’t object always the last word in the phrase, and thus subject to phrase-final lengthening? Could this possibly conceal this difference in this specific context?

9. Please be more clear when describing the statistics in lines 258-69. I.e., please state which variables were continuous, categorical, etc.

Works cited:

Baird, Brandon O. (2014). Acoustic Correlates of Stress in K’ichee’: A Preliminary Investigation. MIT Working Papers in Linguistics, 74: 21-34.

Berinstein, Ava. 1979. A Cross-Linguistic Study on the Perception and Production of Stress. UCLA Working Papers in Phonetics, 47.

England, Nora C. & Brandon O. Baird. (2017). Phonology and Phonetics. In Aissen, Judith, Nora C. England, & Roberto Zavala, (eds.) The Mayan Languages (pp. 175-200). New York: Routledge.

Reviewer 2 Report

I include my comments and suggestions in the enclosed attachment.

Round 2

Reviewer 2 Report

Please, see attachment for specific comments and suggestions.
